# Optimized Properties in Multifunctional Polyphenylene Sulfide Composites via Graphene Nanosheets/Boron Nitride Nanosheets Dual Segregated Structure under High Pressure

**DOI:** 10.3390/nano12193543

**Published:** 2022-10-10

**Authors:** Liangqing Zhang, Shugui Yang, Longgui Peng, Kepeng Zhong, Yanhui Chen

**Affiliations:** 1College of Material Science and Engineering, Xi’an University of Science and Technology, Xi’an 710054, China; 2Shaanxi International Research Center for Soft Matter, State Key Laboratory for Mechanical Behavior of Materials, Xi’an Jiaotong University, Xi’an 710049, China; 3School of Chemistry and Chemical Engineering, Shaanxi Key Laboratory of Macromolecular Science and Technology, Key Laboratory of Special Functional and Smart Polymer Materials, Ministry of Industry and Information Technology, Northwestern Polytechnical University, Xi’an 710072, China

**Keywords:** boron nitride, graphene, polyphenylene sulfide, thermal conductive, electromagnetic interference shielding

## Abstract

The practical application of polymer composites in the electronic and communications industries often requires multi-properties, such as high thermal conductivity (TC), efficient electromagnetic interference (EMI) shielding ability with low electrical conductivity, superior tribological performance, reliable thermal stability and excellent mechanical properties. However, the integration of these mutually exclusive properties is still a challenge, ascribed to their different requirement on the incorporated nanofillers, composite microstructure as well as processing process. Herein, a well-designed boron nitride nanosheet (BN)/graphene nanosheet (GNP)/polyphenylene sulfide (PPS) composite with a dual-segregated structure is fabricated via high-pressure molding. Rather than homogenous mixing of the hybrid fillers, GNP is first coated on PPS particles and followed by encapsulating the conductive GNP layers with insulating BN, forming a BN shell-GNP layer-PPS core composite particles. After hot-pressing, a dual segregated structure is constructed, in which GNP and BN are distinctly separated and arranged in the interfaces of PPS, which on the one hand gives rise to high thermal conductivity, and on the other hand, the aggregated BN layer can act as an “isolation belt” to effectively reduce the electronic transmission. Impressively, high-pressure is loaded and it has a more profound effect on the EMI shielding and thermal conductive properties of PPS composites with a segregated structure than that with homogenous mixed-structure composites. Intriguingly, the synergetic enhancement effect of BN and GNP on both thermal conductive performance and EMI shielding is stimulated by high pressure. Consequently, PPS composites with 30 wt% GNP and 10 wt% BN hot-pressed under 600 MPa present the most superior comprehensive properties with a high TC of 6.4 W/m/K, outstanding EMI SE as high as 70 dB, marvelous tribological performance, reliable thermal stability and satisfactory mechanical properties, which make it promising for application in miniaturized electronic devices in complex environments.

## 1. Introduction

With electronic devices moving toward integration and miniaturization, the inevitability of overheating becomes a serious problem, which results in severe performance deterioration and service life decay of the electronic parts and components [1,2,3]. Along with heat emission, electromagnetic interference (EMI) pollution in modern wireless communication devices is also a problem which not only deteriorates the performance of electronic components but also endangers people’s health [4,5,6]. Moreover, considering the complex application environments of electronic gadgets, electrical insulation, thermal stability, wear resistance and mechanical properties are the prime requirements [7,8]. Therefore, it is attractive to fabricate polymer composites with the integration of these mutually exclusive multifunctional properties, which is meaningful for both the design philosophy itself and the form and capabilities of the products. However, the contradictions between the requirements of incorporated nanofillers, composite microstructure as well as the processing process aimed at different properties make the realization of multifunctionality still challenging. For example, the usage of functional nanofiller is essential in endowing composites with a sort of function, and normally an excess of it is incorporated in polymer composites to ensure a satisfying enhancement in this specific property, while this, in turn, inevitably degrades the inherent mechanical properties of the polymer matrix. Therefore, it is of great importance to ameliorate functional properties under low filler loading [9,10,11]. In pursuit of high-performance multifunctional polymer composites with low filler loading, structure optimization is an effective method. Up to now, extensive investigations into the structural design of polymer composites have been carried out [12]. For instance, sandwich structures [13,14], multilayered structures [15,16], porous structures [17,18], gradient structures [19], and segregated structures [20] were designed, aiming for the restricted distribution and favorable interconnection of functional fillers. Among which, the construction of segregated architecture with conductive fillers selectively located on the boundaries of polymer particles has been widely acknowledged to be an effective method to elevate the EMI shielding and thermal conductive performance. 

Satisfactory EMI shielding of conductive polymer composites (CPC) can be obtained by the construction of a segregated architecture. Yan et al. reported a high-performance electromagnetic interference shielding reduced graphene oxide/polystyrene composite with a multi-facet segregated structure, which exhibits an optimized EMI SE of 45.1 dB [20]. Hu et al. fabricated a segregated polypropylene/ammonium polyphosphate/carbon nanotube (CNT) composite via a microwave-assisted strategy, achieving an EMI SE of 32.1 dB [21]. A comparative study of segregated and homogeneous CNT/polyethylene composites showed that the EMI SE of the former composite was 46% higher than the latter at the same CNT loading [22]. Segregated architecture is also considered an effective approach to construct continuous heat conduction networks. For example, Gao et al. reported a segregated structured BN/ultra-high molecular weight polyethylene (UHMWPE) composite which possesses a high TC of 5.7 W/m/K [23]. Wang et al. also constructed a segregated structure in UHMWPE with a hybrid conductive network, achieving a TC as high as 7.1 W/m/K [24]. Zhou et al. verified the efficient thermal transport along a three-dimensional conductive network in a segregated structure by comparing thermal transport ability with the homogenous composites, and an enhancement of TC from 6.2 to 17.8 W/m/K was achieved [25]. To sum up, it is an effective approach to develop the CPC with desirable thermal conductive or efficient EMI shielding properties by constructing a segregated network structure. 

However, simultaneous amelioration of multifunctional performance is still a challenge in current segregated polymer composites. To our best knowledge, in pursuit of multifunctional properties, most segregated polymer composites are fabricated by incorporating a single nanomaterial with both high thermal and electrical conductivity, such as Ag nanoparticles [7,26], carbon nanotubes [5,27], graphene [28,29,30], and MXene [31,32]. Ascribed to the fact of phonon scattering arising from the high thermal interface resistance between the filler and the polymer matrix, the TC can be significantly improved only at large amounts of filler content, under which a percolative path is formed so that phonons can be transported through the percolated networks. Although a satisfactory EMI shielding ability can be assured, this also leads to ultrahigh electrical conductivity (EC) and a severe impedance mismatch. Thus, the assembly of hybrid fillers with different electrical and thermal properties is necessary to resolve the contradiction of thermal and electrical conductivity. Boron nitride and graphene with high thermal conductivities are effective fillers to upgrade the heat dissipation ability of composites. The incorporation of electrically conductive graphene nanofillers can significantly improve the EMI shielding performance at low filler contents while inescapably increasing the electrical conductivity. Fortunately, boron nitride is an electrical insulator with an intrinsic wide bandgap of ≈5.5 eV. Therefore, the assembly of boron nitride and graphene with a rational structure design may be a promising way to escape the dilemma of electrical and thermal conductive incoordination. 

Besides structural design, the engineering process is also crucial to the ultimate performance of polymer composites. Especially, the loaded pressure is an important processing parameter during conventional polymer processing and several studies have been adopted on pure polymers investigating the effects of pressure on their important properties such as crystallinity, rheology and mechanical properties [33,34,35]. Nevertheless, very little work has addressed the influence of processing pressure on the functionality of polymer composite materials.

In this work, multifunctional performance with the integration of outstanding EMI SE, low electrical conductivity, high TC, excellent thermal stability, enhanced tribological performance and satisfactory mechanical property is realized in BN/GNP/PPS composites. The BN/GNP dual-segregated network structure is first fabricated to resolve the contradiction of thermal and electrical conductivity. In addition, the effects of the mass ratio of BN to GNP and processing pressure on the microstructure and multi-functional performance have been investigated in detail. This work paves the way to fabricate high performance and multifunctional composites.

## 2. Materials and Methods

### 2.1. Materials

A commercial grade PPS with a weight-average molecular mass (Mw) and polydispersity (Mw/Mn) of 4.8 × 10^4^ g/mol and ca. 2, respectively, was synthesized by the Sichuan Deyang Special New Material Co., Ltd (Deyang, China). BN, with the trade name BBN-30, was supplied by the Yaan Bestry Performance Materials Company (Shanghai, China). The average size of BN is approximately 26 μm. The GNP (KNG-182) with an average size of 35 μm was provided by the Xiamen Knano Graphene Technology Co., Ltd (Xiamen, China). All the chemicals used in this work are analytically pure and were used without further purification.

### 2.2. Preparation of BN/GNP/PPS Composites

Fabrication of the BN/GNP/PPS composites with a segregated structure is schematically depicted in Figure 1. BN/GNP/PPS composites with a filler content of 40 wt% were prepared by a two-step solid-state mechanical mixing process and then followed by a high-pressure hot compression shaping process. Initially, PPS particles with an average size of ~900 μm (see Figure 1a) were immersed in ethanol to wet the surface. The treated PPS particles were then mechanically mixed with GNP (Figure 1b) to prepare GNP-coated PPS complex particles. Subsequently, BN (Figure 1c) and GNP/PPS particles were mixed, achieving BN/GNP/PPS particles. The separated mixing processes were both conducted in a mechanical high-speed mixer for 4 min at a speed of 2500 rpm. After that, the BN/GNP/PPS particles were compression molded in a hot press for 480 min, under 0.1, 200, 600 and 1000 MPa, at a temperature of 350 °C, to form cylindrical tablets (with a diameter of~25 mm and a thickness of 3 mm, see Appendix A) for the following characterization. The BN/GNP filler loading was fixed at 40 wt%, while the BN and GNP weight ratios were 0:1, 1:3, 1:1, 3:1 and 1:0, respectively. For comparison, GNP and BN were uniformly mixed with PPS by melt blending using an internal mixer and compression molded under the same condition. For brevity, the ultimate composites with segregated structures were named BN_x_/GNP_y_/PPS and uniformly mixed composites were named BN_x_/GNP_y_/PPS-UM, where x and y represent the weight contents of BN and GNP in the composites.

### 2.3. Characterization

A field emission scanning electron microscopy (SEM, Inspect-F, FEI, Finland) was employed to reveal the freeze-fractured surfaces of PPS composites at an accelerating voltage of 5 kV. The fractured surfaces of the PPS composites were sprayed with gold nanoparticles before SEM observation. The composite sample was initially immersed in liquid nitrogen for 30 min and then quickly fractured. For optical microscopy (OM) measurement, the PPS composite specimens were cut into films of 20 μm thickness using a microtome and observed with an Olympus BX51 polarizing optical microscope (Olympus Co., Tokyo, Japan) with a Micro-Publisher 3.3 RTV CCD camera. The TC of the PPS composites was measured by a Hot Disk instrument (TPS2200, Gothenburg, Sweden) at 25 °C. The sample size for TC measurement was 25.4 mm in diameter and 3 mm in thickness. The electrical resistance of the PPS composites was measured using a Keithley electrometer model 4200-SCS (Beaverton, OR, USA). The samples were cut into rectangular sheets, and the opposites were coated with silver paste to reduce the contact resistance. The electrical conductivity was calculated using the equation σ = L/(S R), where σ and R represent the electrical conductivity and electrical resistance, and L and S are the length and cross-sectional area of the rectangular sheet. EMI shielding measurements were conducted in the frequency range of 8.2−12.4 GHz using an Agilent N5230 network analyzer (Palo Alto, CA, USA). The intermediate frequency bandwidth was set as 1 kHz during the measurement, and 201 points were collected for each specimen. Samples with 25.4 mm in diameter and 3 mm in thickness were placed in the specimen holder, which were connected through the Agilent 85132F coaxial line to separate VNA ports. The scattering parameters (S_11_ and S_21_) of the PPS composites in the frequency range of 8.2–12.4 GHz (X-Band) were gained to calculate the EMI SE. A ball-on-disk frication and wear apparatus (SRV-IV tribo-tester, Optimol Instruments Prüftechnik GmbH, Munich. Germany) was used to measure the coefficient of friction (COF) of the BN/GNP/PPS composites. The ball, with a diameter of 5.0 mm, was made of GCr15 steel. The BN/GNP/PPS composites with a diameter of 25.4 mm and a thickness of 3 mm were fixed on the stage. The experiments were conducted under dry friction conditions (temperature~25 °C, relative humidity ~50%), stroke = 1 mm, normal load = 5 N, duration = 30 min and a sliding frequency 25 Hz. The COF was recorded with a computer. For each experimental condition, each test was carried out three times. The tests on each specimen were re-peated at least three times, after which the average COF was determined. An optical white light interferometer (NPFLEX) was employed to measure the volumes of the wear scar on the BN/GNP/PPS composite disk-like samples. The wear rate was calculated by dividing the wear volume per unit product of sliding distance and load.

## 3. Results and Discussion

### 3.1. Morphological and Structural Analysis

The optical microscopy images of all the PPS composites are shown in Appendix A. Well-designed dual-segregated structures are clearly observed. Considering the similarity of morphologies prepared under different pressures, we only chose composites under 600 MPa as examples for the following discussion. The optical micrographs of the BN/PPS, BN/GNP/PPS, GNP/PPS and BN/GNP/PPS-UM composites processed under 600 MPa are shown in Figure 2. The distribution of BN and GNP in the PPS composites can be easily identified due to the light transmittance difference; that is, the BN enriched region appears to be white, the GNP region appears to be dark, and the PPS appears to be gray. It is obvious that segregated structures exist in the BN/PPS, BN/GNP/PPS and GNP/PPS composites (Figure 2a–e) after compression molding, while BN and GNP are homogenously distributed in the BN/GNP/PPS-UM composite (Figure 2f). Since the filler content is as high as 40 wt%, the filler regions, distributed around the PPS interfaces, are dense and thick enough to form continuous networks. In the filler regions, the GNP and BN fillers are separately distributed rather than homogenously mixed, which is beneficial to a higher TC ascribed to the low phonon scattering at the interfaces of two identical fillers [1]. The rational assembly of BN and GNP in the segregated structure is realized by controlling the coating sequence in the mixing process. GNP is coated on the surface of PPS particles prior to BN, and thus GNP is distributed at the interface between PPS and BN domains. In other words, the GNP layer is coated by BN, and thus one may expect the low electrical conductivity and increased microwave adsorption ability of PPS composites. To observe the detailed microstructure of the interfaces, high-magnification SEM images were further taken, as shown in Figure 3. The interfaces of both PPS and GNP, as well as BN and GNP, are clearly visible in the PPS composites. The PPS particles were not broken up because of the weak flow during hot-pressing. In addition, high processing pressure is also responsible for the distinct interfaces. As depicted in Figure 3g–i, higher magnification images reveal the detailed information about the interfaces of GNP and PPS under different pressures. When compressed under low pressure (0.1 MPa), a relatively ambiguous interface is observed, demonstrating the diffusion of PPS chains into/across GNP particles at a lower pressure. With the increase of compression pressure, distinct stereological textures were observed. With the aid of high pressure, the filler component is squeezed into dense conductive channels, and bundles of GNP were observed to be clustered tightly, which effectively prevents the PPS from penetrating into the conductive filler area. In conclusion, ascribed to the weak flow as well as high processing pressure, the movement of PPS long molecular chains is confined, which would prevent the GNP from penetrating into PPS regions and ensure the selective distribution of GNP.

### 3.2. Thermal Conductive Properties

Figure 4 presents the thermal conductivities of neat PPS, PPS composites with segregated structures, and the BN_10_/GNP_30_/PPS-UM composite, which are fabricated at different pressures. As shown in Figure 4a, neat PPS shows very low TCs, ~0.3 Wm^−1^K^−1^, regardless of the processing pressures. The addition of GNP or BN fillers with inherently high thermal conductivities delivered excellent TC enhancement with respect to that of the pure PPS. For example, the TCs for BN_10_/GNP_30_/PPS-UM composites with randomly distributed fillers increased to approximately 4 Wm^−1^K^−1^, which is more than 10 times compared with neat PPS. More remarkably, the construction of segregated structures in PPS composites further improves the TC; it climbs up to 5~8 Wm^−1^K^−1^ depending on the processing pressure. Take 600 MPa as an example, the TC of BN_10_/GNP_30_/PPS is as high as 6.4 Wm^−1^K^−1^, which is significantly higher than the randomly distributed counterparts (4.8 Wm^−1^K^−1^). The segregated distribution made BN and GNP form percolation so that phonons could be transported through the percolated networks rather than through the BN/PPS or GNP/PPS interfaces where a much higher thermal resistance is imposed. The results reveal that the design of the thermal conductive network in the polymer matrix is crucial for improving the thermal conductivity. 

Another factor affecting TCs of polymer composites that has not been well recognized before is processing pressure. It clearly shows that TCs increase with the augmentation of processing pressure. Why do the TCs of PPS composites increase with the processing pressure? From the above SEM results, one may find that the high pressure has a positive effect on ameliorating the thermally conductive networks to be more compact and improve interfacial bonding between conductive fillers and the PPS matrix, which can effectively reduce the interfacial thermal resistance between filler–filler and filler–matrix. Moreover, the DSC heating curves in Appendix A demonstrate an increase in the crystallinity of the PPS matrix when increasing the pressure, leading to a higher intrinsic TC of the PPS matrix [33]. In general, the increased packing density of fillers in the conductive network improved the thermal conductivity of the PPS matrix as well as reduced interfacial thermal resistance, ultimately ameliorating the TC of PPS composite. Intriguingly, pressure presents a more profound effect on PPS composites with segregated structures than uniformly mixed ones. Take BN_10_/GNP_30_/PPS and BN_10_/GNP_30_/PPS-UM as examples, the TCs of BN_10_/GNP_30_/PPS and BN_10_/GNP_30_/PPS-UM are comparable under 0.1 MPa (4.2 Wm^−1^K^−1^ for BN_10_/GNP_30_/PPS, 4.1 Wm^−1^K^−1^ for BN_10_/GNP_30_/PPS-UM), with the pressure increasing to 1000 MPa, the TC of BN_10_/GNP_30_/PPS sharply increased to 8.5 Wm^−1^K^−1^, while only a small enhancement of 0.2 Wm^−1^K^−1^ is achieved in BN_10_/GNP_30_/PPS-UM. For the BN_10_/GNP_30_/PPS-UM composites, BN and GNP possess a random distribution with few connections to each other. The utilization of high pressure improves the contact possibility of neighboring fillers, reducing the phonon scattering to some extent and slightly improving the TC. While the high pressure is adopted to BN_10_/GNP_30_/PPS with a segregated structure, the fillers are effectively squeezed, which causes efficient phonon transport through the fillers. As the thermal resistance mainly originates from filler interfaces, higher interfacial thermal conductance between BN-BN or GNP-GNP can ultimately increase the thermal conductivities of PPS composites with segregated structures. Furthermore, by assembling GNP and BN fillers into the segregated structure, the thermal conductivities of PPS composites were further improved. Figure 4b clarifies how the assemble ratio affects the TCs. In comparison with BN, GNP has higher intrinsic TC. Keeping the total filler content fixed at 40 wt%, the TCs for composites under 600 MPa continuously climb up from 3.5 Wm^−1^K^−1^ to 6.8 Wm^−1^K^−1^ with the increased GNP content. It is worth noting that the synergistic enhancement effect of BN and GNP is stimulated under high pressure. For example, the TC for BN_10_/GNP_30_/PPS is comparable with that of GNP_40_/PPS under 600 MPa, and it far outstrips the value of GNP_40_/PPS when the pressure increases to 1000 MPa, reaching a conductivity of 8.5 Wm^−1^K^−1^, which is 25% and 136% higher than GNP_40_/PPS and BN_40_/PPS. It is speculated that the intercalation structure is formed under high pressure and the interfacial thermal resistance between the neighboring GNP is effectively reduced by the insertion of lamellar BN. In general, the combination of segregated structures, the rational assemble of GNP with BN and high processing pressure have a synergetic effect on attaining a high TC for PPS composite. It is worth mentioning that although the TC has been elevated to a large extent compared with pure PPS, it is still far lower than the intrinsic TC of both BN and GNP, which is ascribed to the limitation of filler content and thermal interface resistance. With increasing filler content, the TC of PPS composites could continuously be further improved, but a high filler loading could compromise mechanical flexibility and processibility. Therefore, the filler content in PPS composites should be maintained to achieve a balance between thermal conductivity and the desired multifunctional properties. On the other hand, interfacial thermal resistance is an important part of influencing the TC of PPS composites, which includes those between BN and BN, GNP and GNP, BN and GNP, BN and PPS matrix and GNP and PPS matrix. Efforts can be made to reduce the phonon scattering at the interfaces, such as the construction of percolative paths and the rational distribution of BN and GNP. Nevertheless, heat loss is inevitable as long as there are interfaces.

To visualize this heat dissipation difference, infrared imaging was used to record the temperature response during heating, as shown in Figure 5. We can clearly see that the surface temperatures of PPS composites with segregated structured conductive networks rise much faster than the uniformly mixed PPS composite and neat PPS, following the sequence: GNP_40_/PPS > BN_10_/GNP_30_/PPS > BN_40_/PPS > BN_10_/GNP_30_/PPS-UM > PPS. This transient temperature response agrees well with the thermal conductivity (Figure 4). The surface temperatures of BN_10_/GNP_30_/PPS, BN_10_/GNP_30_/PPS-UM and PPS were scanned from the bottom along the white dotted line (see Figure 5a), as plotted in Figure 5b. The BN_10_/GNP_30_/PPS presents a more homogenous temperature distribution compared with the BN_10_/GNP_30_/PPS-UM and PPS, that is, a relatively lower temperature near the heating source while there is a higher temperature far from the heating source, indicating better thermal diffusion and dissipation capacity. Overall, the conclusion can be brought to light that BN/GNP/PPS composites with dual-segregated structures processed under high pressure allow for efficient heat transferring and dissipating, showing a bright prospect as the thermal conductive media for next-generation microelectronic devices. 

### 3.3. Electrical Conductivity and EMI Shielding Performance of PPS Composites

The electrical conductivity of PPS composites as a function of pressure is shown in Figure 6. The electrical conductivities of neat PPS and BN_40_/PPS composites are kept constantly at ~10^−13^ S/m, ascribed to the insulation property of PPS and BN. With the introduction of GNP content, the electrical conductivity increased monotonically. In addition, the segregated structured composites present higher electrical conductivity and increase in the wake of ascending pressure, while the uniformly mixed composites and neat PPS are stubborn to pressure. For example, the electrical conductivity for BN_30_/GNP_10_/PPS display a significant increase of approximately 2 orders of magnitude (from 6.7 × 10^−4^ S/m to 0.019 S/m) when the pressure rises from 0.1 MPa to 1000 MPa. Given the same filler content, an increment of 51% from 0.055 S/m to 0.083 S/m is achieved in the BN_10_/GNP_30_/PPS composites, while there is no obvious change for BN_10_/GNP_30_/PPS-UM with the increase in processing pressure. Despite the compression of the conductive network at high pressure, the electrical conductivity is still far lower than traditional segregated structured EMI-shielding materials (Appendix A), which is ascribed to the insulating BN layers that break the GNP conducting network. 

It is well acknowledged that electrical conductivity is one of the critical parameters affecting electromagnetic interference shielding. Surprisingly, despite the relatively low electrical conductivity, satisfying electromagnetic interference shielding effectiveness (EMI SE) is achieved in PPS composites, as shown in Figure 7. With 10 wt% GNP loading, the EMI SE surpasses the standard value for commercial applications (20 dB), and the increasing GNP content directly causes higher EMI SE. The maximum EMI SE at 30 wt% GNP is 70 dB, which is quite a high level compared with previously reported works (as summarized in Appendix A). It is worth mentioning that all of the efforts made in this work, including the construction of segregated conductive network, the adoption of high pressure and the rational assembly of BN and GNP, contribute to the excellent EMI shielding performance. Sharing the same component composition, the BN_10_/GNP_30_/PPS composite featuring a segregated structure presents a superb SE of 70 dB, while a lower value of 54 dB is achieved for the BN_10_/GNP_30_/PPS-UM composite in which fillers are uniformly distributed, indicating the ascendency of the segregated structure on EMI shielding. The localized distribution of GNP in a segregated structure can greatly enhance the utilization ratio, leading to increased electrical conductivity of the composite. More importantly, the densely packed GNP networks at the interfaces can effectively interact with incident radiation, leading to the very high EMI SE. Furthermore, Appendix A gives a comparison of the EMI SE of composites with segregated structures, in which our data outperforms most previously published values. The superiority of BN/GNP/PPS composites constructed in our work may originate from the adoption of high pressure during the hot-pressing process. First of all, high pressure can directly contribute to more continuous and denser conductive networks, which is favorable for the sufficient interaction of interconnected GNPs with incoming electromagnetic waves, accordingly, leading to higher EMI SE. As verified in Figure 7b, with the increase of processing pressure, the SE for BN_10_/GNP_30_/ PPS composite climbs sharply from 47.7 dB for 0.1 MPa to 67.8 dB for 1000 MPa. In addition, it is well known that the EMI SE of a shielding material is considerably pertinent to the filler content. Nevertheless, ascribed to the solidity and nondeformability of functional fillers, too much of them is adverse to the adhesion of polymer particles in a segregated structure. Applying high pressure can effectively help escape from the dilemma by strengthening the GNP-BN as well as GNP-PPS binding forces, and the filler content in this work can be as high as 40 wt%. Consequently, more conductive fillers participate in blocking incidental electromagnetic waves and directly result in higher EMI SE. It is noticeable that the EMI SE of BN_10_/GNP_30_/PPS is even higher than that of GNP_40_/PPS, thus we comment that synergistic enhancement in the EMI SE was achieved by the BN and GNP dual segregated structure.

In order to reveal the shielding mechanism of PPS composites, total SE (SE_T_), microwave reflection SE (SE_R_) and microwave absorption (SE_A_) of PPS composites are illustrated in Figure 7c,d. The SE_T_, SE_A_ and SE_R_ of BN_10_/GNP_30_/PPS under 600 MPa are 67.4 dB, 60.5 dB and 6.9 dB, respectively. For comparison, the values of the BN_10_/GNP_30_/PPS-UM and GNP_40_/PPS are also presented. GNP_40_/PPS presents the highest SE_R_, which is consistent with conductivities and attributed to the improved impedance mismatch between shielding surface and air. However, the SE_A_ of BN_10_/GNP_30_/PPS is superior to both BN_10_/GNP_30_/PPS-UM and GNP_40_/PPS. This implies that the superior EMI shielding performance of BN_10_/GNP_30_/PPS can be attributed to the synergistic effect of the segregated structure and the hybrid of conductive fillers. By constructing a segregated structure, well developed conducting networks are formed, which is favorable for the ohmic loss and the special architecture provides many interfaces to reflect, scatter and absorb incident microwaves in the segregated unit cells. Moreover, as speculated in the former part, intercalation structure is formed under high pressure, thus the insertion of BN nanosheets into the GNP conductive network could generate more interfaces and act as defects. Owing to the large contrast in electrical conductivity between BN and GNP, free space charges can spontaneously accumulate at the interfaces under the electromagnetic field, which will induce the interfacial polarization effect and dipole polarization effect. It is worth noting that although SE_A_ is far higher than SE_R_, this does not mean that the shielding mechanism is adsorption dominated, since SE_A_ and SE_R_ only represent the ability of absorption loss and reflection loss [36]. As the processing pressure increases, the reflection effectiveness is almost unchanged, while the average SE_T_ was significantly promoted from 47.7 dB for 0.1 MPa to 67.8 dB for 1000 MPa. Thus, the enhanced EMI shielding performance is mainly attributed to the enhanced adsorption loss. Such enhancement indicates that the adoption of high pressure could significantly promote the SE_A_ by conductive loss, which is closely related to the interconnected GNP conductive networks. Specifically, the conducting network becomes more perfect after high pressure loading and could dissipate more electromagnetic energy by conductive loss. Meanwhile, the distinct interfaces of GNP-BN in the intercalation structure formed under high pressure can effectively initiate the accumulation of charge at the heterogeneous interfaces, resulting in enhanced interfacial polarization loss. In general, excellent EMI shielding performance with low electrical conductivity and decreased SE_R_ is realized by rationally assembling GNP with BN in a segregated structured composite under high pressure_._

It has become a critical design factor for miniaturized electronic components to incorporate EMI shielding and heat dissipation into the same part. Therefore, materials simultaneously endowing high EMI SE and TC are urgently needed. Figure 8a presents a comparison of EMI SE and TC obtained in this work and previously reported in the literature (detailed information can be seen in Appendix A). Encouragingly, our fabricated BN/GNP/PPS composites simultaneously achieved superior EMI SE and TC. 

Schematic illustrations of thermal conduction and the EMI shielding mechanism are shown in Figure 8b. The synchronous realization of satisfying thermal conductive and EMI shielding performance is attributed to the well-defined segregated structure. Since both GNP and BN have excellent thermal conductivity, a perfect 3D thermal conductive path is formed by GNP and BN dual-segregated structures, which provides an expressway for phonon conduction. In addition, GNP and BN are separately distributed in the interface of PPS, which is favorable for minimizing the interfacial thermal resistance between GNP and BN fillers. Moreover, the application of high pressure can effectively improve the interfacial bonding and release the interstice inside composites, greatly reducing the phone scattering at interfaces or void defects. 

The dual-segregated structures also play a crucial role in EMI shielding. When microwaves are incident on the surface of PPS composites, some of them are reflected and residual enters the material. Then, incident microwaves are trapped inside the segregated unit cells and dissipated in the form of heat by multiple reflection and scattering. At the same time, the heat transformed from incident microwaves could be dissipated quickly through conductive networks. Therefore, the BN/GNP/PPS composites with superior EMI shielding performance and excellent thermal management performance are expected to find a wide range of potential applications in the fields of the communication industry, artificial intelligence and wearable electronics.

### 3.4. Tribological, Mechanical and Thermal Properties

Aside from the outstanding thermal conductivity and the excellent EMI shielding properties, we also investigate the wear-resisting performance, mechanical properties and thermal stability of PPS composites. Good wear resistance is important for a material applied in a harsh environment. Hence, the tribological properties of these PPS composites were studied by using a ball-on-disk frication and wear apparatus. Two main performance indicators of COF and wear rate are shown in Figure 9. The homogenous mixed PPS composites show lower COF than neat PPS because the layer structures of both GNP and BN can work together with the PPS matrix and form a stable lubricating layer on the wear surfaces, which has been verified by the OM observations of the wear surfaces (Appendix A). Intriguingly, the COF of PPS composites with segregated structures further decreases by 50% compared with the uniformly-mixed PPS composites, which is as low as 0.03 for the BN_20_/GNP_20_/PPS composite processed under 0.1 MPa, indicating that the segregation between layer-structured filler and the PPS matrix is able to attain good lubrication effect. The improved wear resistance performance of the segregated PPS composite mainly comes from the good thermal conductivity and the densely packed BN and GNP layers. During the fraction process, an instantaneous high temperature is generated and softens the surface of PPS composites. Therefore, it is easy for the steel ball to make the friction surface rough, leading to a large friction coefficient. In the case of BN/GNP/PPS composites, the GNP and BN dual-segregated structure with excellent thermal conductivity can transform the heat immediately, which is effective to elevate the friction performance. In addition, a small amount of BN and GNP will migrate out from the densely-packed networks and form a transfer film between the steel ball and the wear surface, which can provide reliable protection for the PPS matrix, thus decreasing the surface roughness and friction coefficient of the friction surface. However, an excessive amount of BN and GNP may exfoliate from the friction surface in the homogenous mixed PPS composites, which inevitably causes an increased friction coefficient. Both the PPS composites and neat PPS processed under 0.1 MPa and 200 MPa show a lower COF than those processed under 600 MPa and 1000 MPa. The reason for the increased COF of PPS samples processed under 600 MPa and 1000 MPa is likely to be caused by the increased crystallinity of PPS. As shown in Appendix A, the crystallinity of both neat PPS and PPS composites increases from 50% for 0.1 MPa and 200 MPa to 80% for 600 MPa and 1000 MPa, resulting in the less flexible PPS matrix to form a lubricating layer. However, the negative effect of high pressure on BN_10_/GNP_30_/PPS is negligible, which presents a COF as low as 0.05 under 600 MPa and 1000 MPa. Figure 9b shows the wear rate of both PPS composites and neat PPS. Even though the COF varied with the loading of fillers and the segregated structures, it is clear that the wear rates of these PPS samples are, in fact, at the same level (~10^−5^ mm^3^N^−1^m^−1^), which is in the range of the reported wear rate of PPS. This may be due to the similar wear mechanism, that is, the adhesive–abrasive mechanism of all the PPS composite samples, as the wear surfaces show adhesion and micro-cutting (Appendix A).

In addition to wear resistance, mechanical properties are also essential for practical application. It is reported that conventional segregated structured composites fabricated by solid mixing and thermoforming technology always present seriously deteriorated mechanical properties due to the weak interaction between the polymer matrix and conductive fillers. In the current work, with the filler content as high as 40 wt%, only a slight decrease in the compressive strength for segregated PPS composites is observed, which can reach 146 MPa (Appendix A), outperforming most segregated polymer composites.

The thermal stability of PPS composites was also investigated. The TGA curves of neat PPS and composites are shown in Appendix A. Both neat PPS and composites maintain their stability up to ~480 °C and are pyrolyzed in the temperature range of 500~600 °C. The addition of conductive fillers can further increase the thermal degradation temperature ascribed to the thermal protection effect of GNP and BN. No significant influence of the processing pressures on the thermal stability was observed. These results indicate that the PPS composites inherit the excellent thermal stability of PPS resin.

Overall, the composites not only present simultaneously excellent thermal conductive and EMI shielding performance but also multi-functional performance by integrating good wear resistance performance, mechanical properties and thermal stability.

## 4. Conclusions

In summary, multifunctional PPS composites have been fabricated under the strategies of dual segregated structure construction and high-pressure molding. Efficient thermal transport is realized in segregated PPS composites, reaching a thermal conductivity as high as 8.5 W/m/K. Meanwhile, the electrical conductivity is significantly decreased, which is much lower than previously reported segregated polymer composites with a single filler. The segregated dual network structure and high pressure are also favorable for EMI performance by creating numerous interfaces for microwaves, reflecting scattering and absorption, and a remarkable EMI SE of 70 dB is achieved. Moreover, the composites also have an excellent anti-fraction ability with a COF of 0.03, mechanical properties with a compressive strength as high as 146 MPa, and thermal stability up to 480 °C. The excellent comprehensive properties make it promising for PPS composites applied in new generation portable and miniaturized electronic devices.

## Figures and Tables

**Figure 1 nanomaterials-12-03543-f001:**
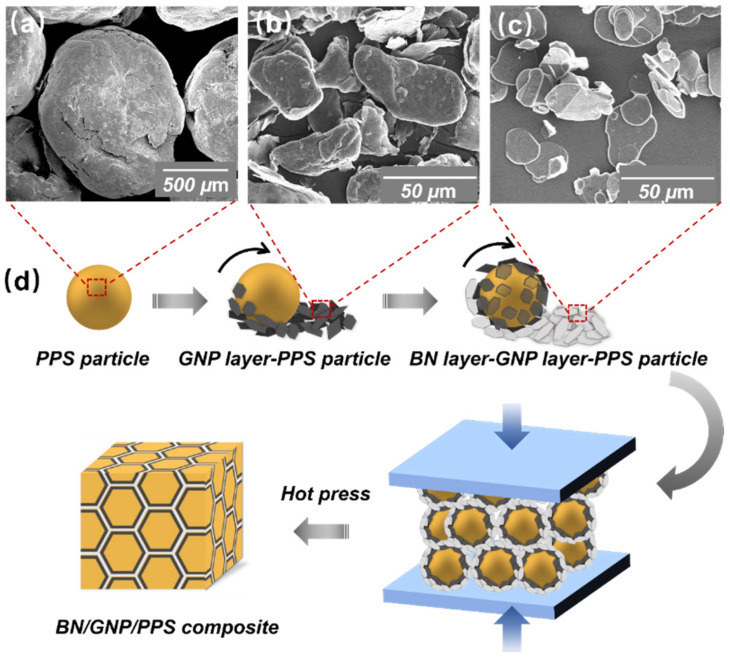
SEM images of (**a**) PPS particle, (**b**) GNP and (**c**) BN; (**d**) Schematic representation of fabrication process of BN/GNP/PPS composites with dual segregated structure.

**Figure 2 nanomaterials-12-03543-f002:**
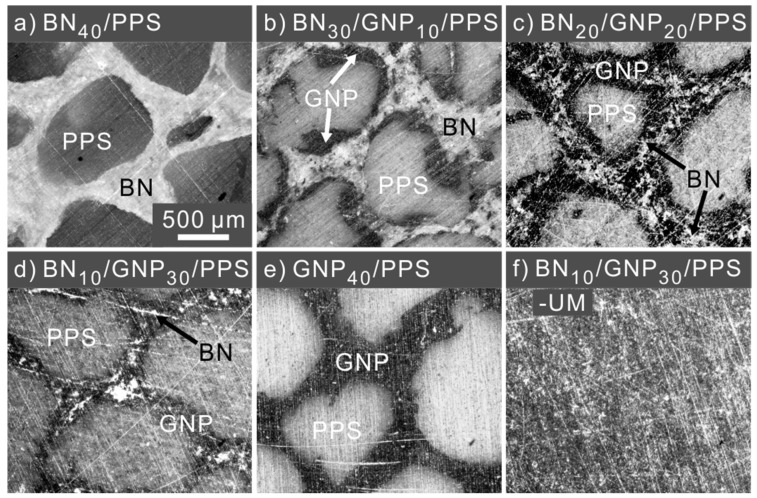
OM images of (**a**) BN_40_/PPS, (**b**) BN_30_/GNP_10_/PPS, (**c**) BN_20_/GNP_20_/PPS, (**d**) BN_10_/GNP_30_/PPS, (**e**) GNP_40_/PPS and (**f**) BN_10_/GNP_30_/PPS-UM composites processed under 600 MPa.

**Figure 3 nanomaterials-12-03543-f003:**
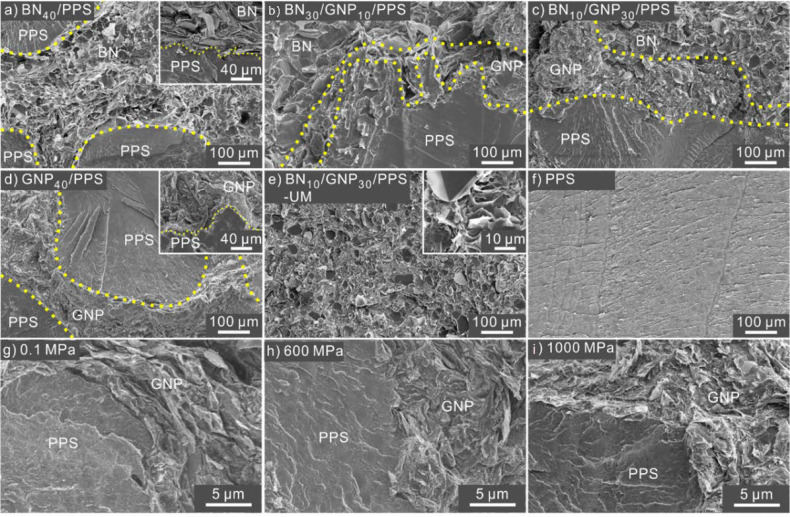
SEM images of PPS composites processed under 600 MPa (**a**–**f**) and GNP_40_/PPS composite under different pressures (**g**–**i**).

**Figure 4 nanomaterials-12-03543-f004:**
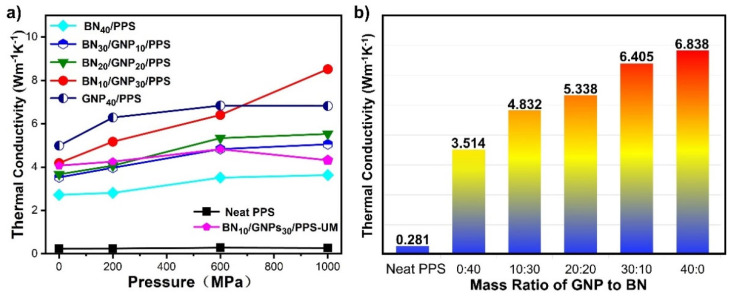
(**a**) Thermal conductivity of PPS composites with segregated structures (BN_40_/PPS, BN_30_/GNP_10_/PPS, BN_20_/GNP_20_/PPS, BN_10_/GNP_30_/PPS, GNP_40_/PPS), neat PPS and BN_10_/GNP_30_/PPS composites with uniformly-distributed fillers, which were fabricated at different pressures; (**b**) Thermal conductivity of neat PPS and PPS composites with different mass ration of GNP to BN, which were processed under 600 MPa.

**Figure 5 nanomaterials-12-03543-f005:**
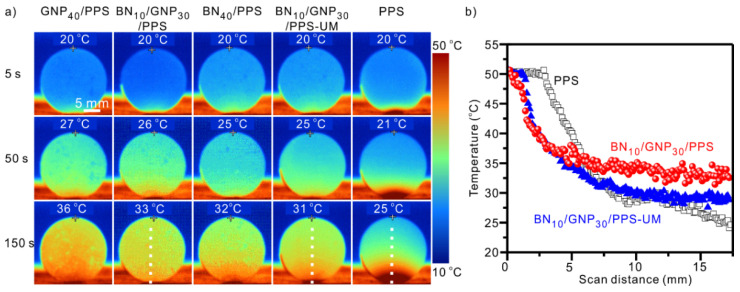
(**a**) Infrared thermal images of PPS and PPS composites processed under 600 MPa during the heating process and (**b**) surface temperature variation as a function of heating distance.

**Figure 6 nanomaterials-12-03543-f006:**
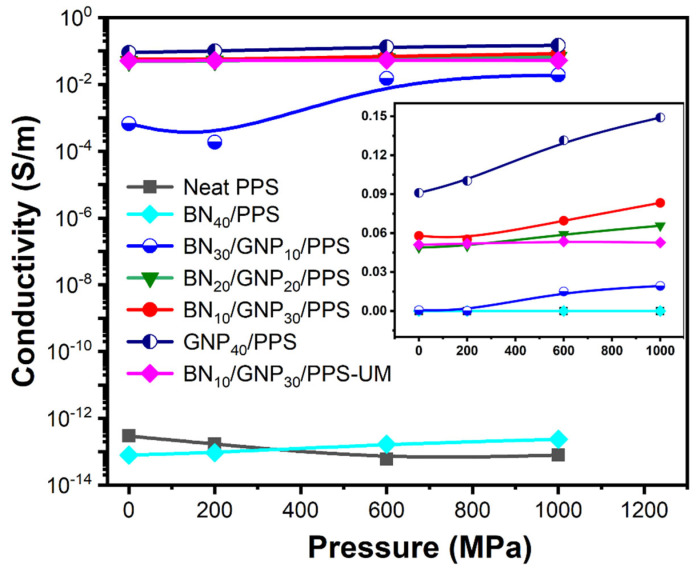
Electrical conductivities of PPS composites with segregated structures (BN_40_/PPS, BN_30_/GNP_10_/PPS, BN_20_/GNP_20_/PPS, BN_10_/GNP_30_/PPS, GNP_40_/PPS), neat PPS and BN_10_/GNP_30_/ PPS-UM composites and inset is the conductivities shown on a linear scale.

**Figure 7 nanomaterials-12-03543-f007:**
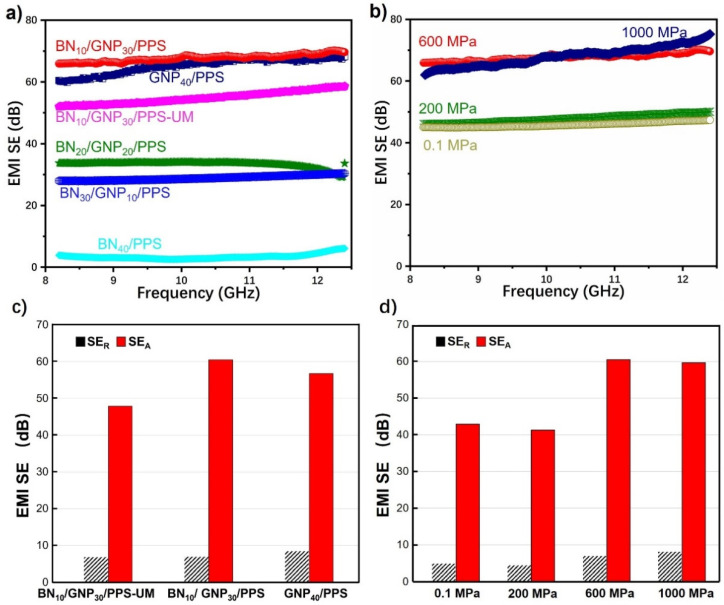
EMI shielding effectiveness of (**a**) PPS composites with segregated structures (BN_40_/PPS, BN_30_/GNP_10_/PPS, BN_20_/GNP_20_/PPS, BN_10_/GNP_30_/PPS, GNP_40_/PPS), neat PPS and BN_10_/GNP_30_/PPS-UM composites with uniformly-distributed fillers fabricated under 600 MPa, (**b**) BN_30_/GNP_10_/PPS composites fabricated under different pressures, (**c**) SE_T_, SE_A_ and SE_R_ of BN_10_/GNP_30_/PPS-UM, BN_10_/GNP_30_/PPS and GNP_40_/PPS composites fabricated under 600 MPa, and (**d**) SE_T_, SE_A_ and SE_R_ of BN_10_/GNP_30_/PPS composites as a function of pressure.

**Figure 8 nanomaterials-12-03543-f008:**
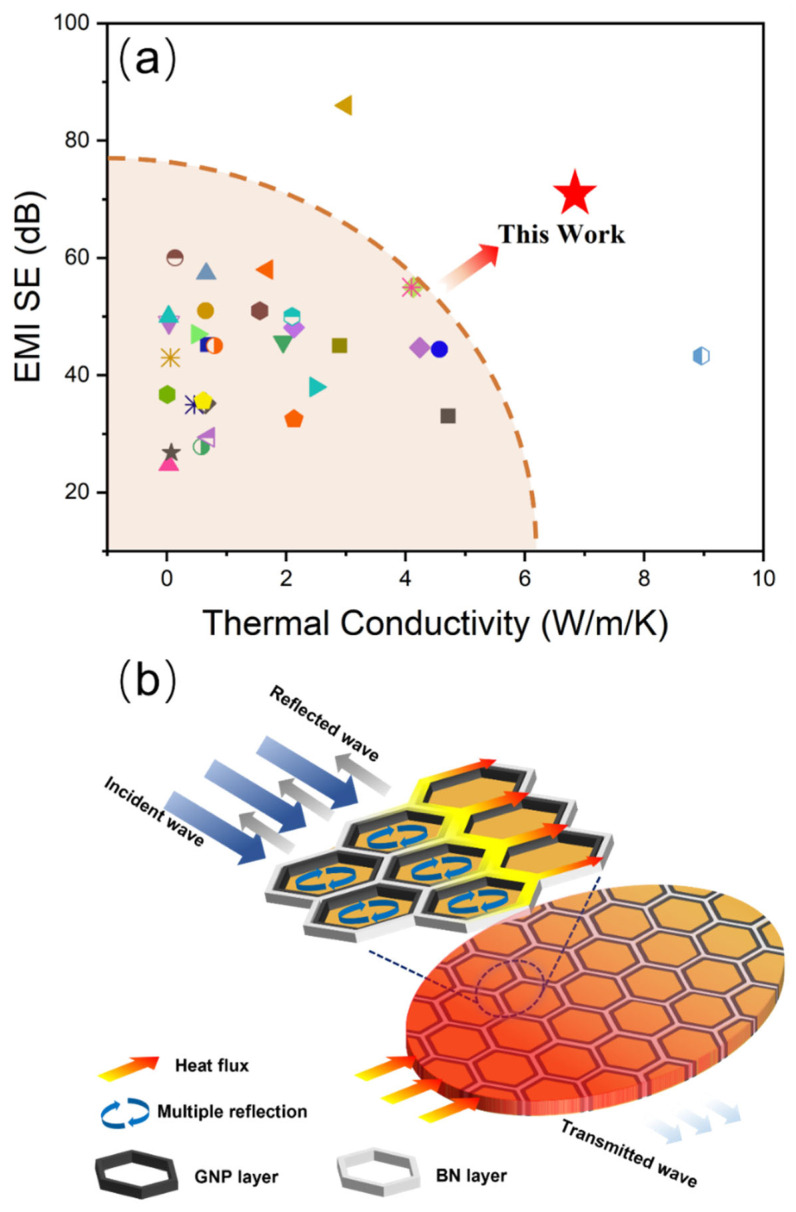
(**a**) Comparison of EMI SE and thermal conductivity of BN/GNP/PPS composites with other reported composites (detailed information in Appendix A), (**b**) Schematic illustration of the EMI shielding mechanism for BN/GNP/PPS composites with segregated structure.

**Figure 9 nanomaterials-12-03543-f009:**
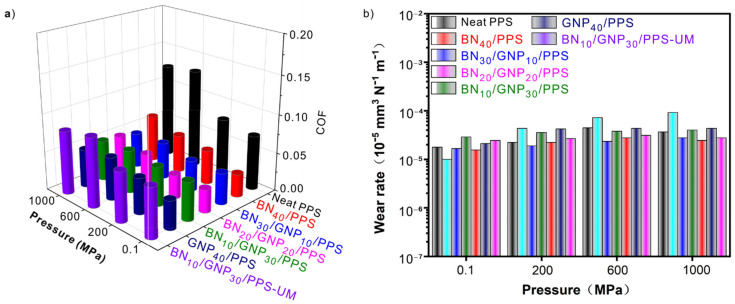
The tribological performance of PPS composites with segregated structures (BN_40_/PPS, BN_30_/GNP_10_/PPS, BN_20_/GNP_20_/PPS, BN_10_/GNP_30_/PPS, GNP_40_/PPS), neat PPS and BN_10_/GNP_30_/PPS-UM composites with uniformly-distributed fillers, which were fabricated at different pressures. (**a**) COF, (**b**) Wear rate.

## Data Availability

The data presented in this study are available on request from the corresponding author.

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
