# Peer review of "Optimized Properties in Multifunctional Polyphenylene Sulfide Composites via Graphene Nanosheets/Boron Nitride Nanosheets Dual Segregated Structure under High Pressure"

_nanomaterials, 2022, doi:10.3390/nano12193543_

Round 1

Reviewer 1 Report

In this paper, the authors present a boron nitride nanosheets (BN)/graphene nanosheets (GNP)/polyphenylene sulfide (PPS) composite. The GNP is first coated on PPS particles and then followed by encapsulation with insulating BN, forming a dual segregated structure that exhibits high thermal conductivity and efficient electromagnetic interference (EMI) shielding ability.

The application is original. The paper is clear and well organized. The manuscript is detailed and reports a well-performed experimental work that is satisfactorily analyzed.

The manuscript can be suitable for publication after a revision. Here are points that the authors should consider:

-        For example, the usage of functional nanofiller is essential in endowing composites with a sort of function and normally excessive of it is incorporated in polymer composite to ensure a satisfying enhancement in this specific property, while this in turn inevitably degrades the inherent mechanical properties of polymer matrix. Therefore, it is of great importance to ameliorate functional properties under low filler loading.” There are plenty of works on polymers loaded with a variety of nanofillers. The authors could add a few citations here (example: https://doi.org/10.1016/j.compositesb.2017.10.020)

-        Satisfactory EMI shielding of CPC can be obtained by the construction of segregated  architecture.” Define CPC acronym.

-        “Ascribed to the fact of phonon scattering arising from the high thermal interface resistance between the filler and polymer matrix, the TC can be significantly improved only at large amounts of filler contents.” Complete the sentence clarifying why large amounts of filler are required for high TC. Is this because percolative paths are necessary?

-        “The samples were cut into rectangular sheets, and the opposites 161 were coated with silver paste to eliminate the contact resistance.” Silver paste can reduce but not eliminate the contact resistance.

-        “The electrical conductivities of neat PPS and BN40/PPS composite kept constantly 306 at ~10^-13 S/m, ascribed to the insulation property of PPS and BN.” It might be convenient adding an inset in Figure 6 in which the conductivities are shown on a logarithmic scale.

-        In the introduction it is said: “The incorporation of electrically conductive graphene nanofillers can significantly improve the EMI shielding performance at low filler contents, while inescapably increase the electrical conductivity. Fortunately, boron nitride is an electrical insulator with an intrinsic wide bandgap of ≈5.5 eV. Therefore, the assemble of boron nitride and graphene with rational structure design may be a promising way to escape the dilemma of electrical and thermal conductive incoordination.” This paragraph seems to contradict the statement in section 3.3. “In general, high electrical conductivity 325 contribute to achieving excellent EMI shielding performance by enhancing EM reflection 326 and adsorption and the target conductivity for EMI application is 1 S/m.[7]” Please clarify.

Author Response

Attached file is the respons to reviwer 1

Reviewer 2 Report

Zhang and his co-workers present in this paper the fabrication of a boron nitride nanosheets (BN)/graphene nanosheets (GNP)/polyphenylene sulfide (PPS) composite with dual segregated structure via high-pressure molding. Component ratio and processing pressure have been studied to obtain the best performing material. Composites were characterized by various techniques, including SEM and optical microscopy, to determine the morphology, thermal conductivity, electrical resistance, efficient electromagnetic interference shielding, thermal stability, wear and friction coefficient of synthesized samples. The research is carefully done, and the manuscript is well-written. A final reading of the manuscript is necessary to eliminate typing mistakes. Authors should take care about formatting issues, namely, that literature reference should come before the ‘,’ or “.”, like “[10,11],” or “[7,8].”  Please correct throughout the manuscript. 

Author Response

Attached file is the respons to reviewer 2

Reviewer 3 Report

The following issues must be addressed:

1.       I recommend to change the word “ameliorated” with “Optimized” in the manuscript title;

2.       The last paragraph in the Introduction should be improved in order to outline what is new and innovative compared with other similar work;

3.       Please explain in more details why the separately distributed GNP and BN fillers are better than homogenously mixed;

4.       Which are the design limitations on improving the thermal conductivity of the samples? Explain in the manuscript;

5.       There are specific parameters that can be tailored to improve the structure segregation;

6.       Conclusion part should be improved to underline the most representative results.

Author Response

Attached file is the respons to reviewe 3

Round 2

Reviewer 3 Report

The manuscript can be published in present form.